# Nanosecond pulsed electric fields induce the integrated stress response via reactive oxygen species-mediated heme-regulated inhibitor (HRI) activation

Yoshimasa Hamada[1,2], Yuji Furumoto[3], Akira Izutani[3], Shusuke Taniuchi[1,2,4], Masato Miyake[1,2,4], Miho Oyadomari[1], Kenji Teranishi[3], Naoyuki Shimomura[3], Seiichi Oyadomari[1,2,4] *

1 Division of Molecular Biology, Institute for Genome Research, Institute of Advanced Medical Sciences, Tokushima University, Tokushima, Japan, 2 Fujii Memorial Institute of Medical Sciences, Institute of Advanced Medical Sciences, Tokushima University, Tokushima, Japan, 3 Institute of Technology and Science, Tokushima University, Tokushima, Japan, 4 Department of Molecular Physiology, Diabetes Therapeutics and Research Center, Institute of Advanced Medical Sciences, Tokushima University, Tokushima, Japan

* oyadomar@tokushima-u.ac.jp

**Data Availability Statement:** All relevant data are within the manuscript and its Supporting Information files.

## Abstract

The integrated stress response (ISR) is one of the most important cytoprotective mechanisms and is integrated by phosphorylation of the α subunit of eukaryotic translation initiation factor 2 (eIF2α). Four eIF2α kinases, heme-regulated inhibitor (HRI), double-stranded RNA-dependent protein kinase (PKR), PKR-like endoplasmic reticulum kinase (PERK), and general control nonderepressible 2 (GCN2), are activated in response to several stress conditions. We previously reported that nanosecond pulsed electric fields (nsPEFs) are a potential therapeutic tool for ISR activation. In this study, we examined which eIF2α kinase is activated by nsPEF treatment. To assess the responsible eIF2α kinase, we used previously established eIF2α kinase quadruple knockout (4KO) and single eIF2α kinase-rescued 4KO mouse embryonic fibroblast (MEF) cells. nsPEFs 70 ns in duration with 30 kV/cm electric fields caused eIF2α phosphorylation in wild-type (WT) MEF cells. On the other hand, nsPEF-induced eIF2α phosphorylation was completely abolished in 4KO MEF cells and was recovered by HRI overexpression. CM-H$_2$DCFDA staining showed that nsPEFs generated reactive oxygen species (ROS), which activated HRI. nsPEF-induced eIF2α phosphorylation was blocked by treatment with the ROS scavenger N-acetyl-L-cysteine (NAC). Our results indicate that the eIF2α kinase HRI is responsible for nsPEF-induced ISR activation and is activated by nsPEF-generated ROS.

## Introduction

The integrated stress response (ISR) is one of the most important cytoprotective mechanisms and is integrated by phosphorylation of the α subunit of eukaryotic translation initiation factor

**Funding:** This work was supported by JSPS KAKENHI Grant Number JP19H0285310 (S.O.).

**Competing interests:** The authors have declared that no competing interests exist.

2 (eIF2α) at Ser51. The ISR is activated in response to several stress conditions, such as viral infection, heme deprivation, amino acid starvation, reactive oxygen species (ROS) and endoplasmic reticulum (ER) stress [1, 2]. The phosphorylation of eIF2α results in the inhibition of eIF2-GTP/Met-tRNAi ternary complex recycling, which is necessary for the initiation of mRNA translation, thereby reducing overall translation while selectively favoring the translation of proteins implicated in stress recovery, such as activating transcriptional factor 4 (ATF4) [3, 4]. ATF4 induces the expression of several genes involved in the regulation of redox balance, amino acid biosynthesis and transport to overcome the imposed stress and restore cellular homeostasis. There are four different eIF2α-specific kinases, namely, heme-regulated inhibitor (HRI), protein kinase double-stranded RNA-dependent (PKR), general control nonderepressible (GCN) 2, and PKR-like ER kinase (PERK), and each eIF2α kinase senses and responds to distinct cellular stresses, with some overlap in their activities [5].

As the ISR is an innate protective mechanism, dysregulation of ISR signaling has important pathologic consequences linked to inflammation [6], diabetes [7], cancer [8], and neurodegenerative diseases [9]. Furthermore, enhancement of ISR signaling has been suggested to have beneficial effects, further supported by genetic manipulation of the ISR pathway in mouse models of neurodegenerative diseases such as multiple sclerosis [10] and amyotrophic lateral sclerosis [9]. Hence, modulation of the ISR represents a promising therapeutic strategy, and recent encouraging advances have been made in this area through the development of small molecules to enhance ISR signaling. Indeed, several compounds have been described to activate the eIF2α kinases, 1H-benzimidazole-1-ethanol,2,3-dihydro-2-imino-a-(phenoxy-methyl)-3-(phenylmethyl)-,monohydrochloride (BEPP) as a PKR activator [11], 1-(benzo[d][1,2,3]thiadiazol-6-yl)-3-(3,4-dichlorophenyl)urea (BTdCPU) as an HRI activator [12], halofuginone as a GCN2 activator [13] and CCT020312 as a PERK activator [14]. Halofuginone, for example, is a potent inhibitor of angiogenesis progression and is being evaluated in a clinical phase II trial [15].

Physiotherapy, such as electrotherapy, thermotherapy and phototherapy, has a place within clinical practice. Different modalities of therapy that activate the ISR in addition to drug therapy are worth developing. We previously reported that nanosecond pulsed electric fields (nsPEFs) induce eIF2α phosphorylation [16]. nsPEFs are characterized by ultrashort-duration and high-intensity electric fields [17]. Typical nsPEFs have a duration of 60–300 ns, with a rise time of 4–30 ns [18, 19]. Millisecond PEFs are commonly used in life sciences, especially for DNA transfection to generate pores on the cell membrane. On the other hand, nsPEFs can directly reach intracellular components without cell membrane destruction, and thus nsPEFs have emerged as a unique therapeutic tool for intracellular manipulation without any chemical intervention [20, 21]. Although nsPEFs are now recognized as a drug-free and purely electrical cancer therapy, the molecular mechanism of nsPEF action remains largely unclear.

In this study, we investigated which eIF2α kinase is responsible for nsPEF-induced eIF2α phosphorylation and how nsPEFs activate the responsible eIF2α kinase. Here, we present evidence that nsPEFs generate ROS that activate HRI, leading to eIF2α phosphorylation. Our results provide a molecular mechanism for the action of nsPEFs for research and therapeutic development.

## Materials and methods

### Cell culture and cell lines

SV40 large T-antigen immortalized mouse embryonic fibroblasts (MEFs) were cultured in DMEM-high glucose supplemented with 10% FBS (Giboco), 2 mM L-glutamine (Nakarai-Tesque, Japan), 55 μM 2-mercaptoethanol, and nonessential amino acids (Invitrogen) at 37°C

under humidified conditions with 5% $CO_2$. eIF2α kinase quadruple knockout (4KO) and single eIF2α kinase-rescued 4KO MEF cells were previously established [22]. Wild type, *HRI* KO, and *GCN2* KO in Hap1 cells were cultured in IMDM (HyClone) supplemented with 10% FBS, 55 µM 2-mercaptoethanol (Invitrogen) at 37˚C under a humidified condition with 5% CO2. *HRI* KO and *GCN2* KO in Hap1 cells were generated by CRISPR/Cas9 system.

### Electrical devices for the generation of nsPEFs

A pulsed power generator, based on a Blumlein pulse-forming network (B-PFN) that generates nsPEFs, was designed and developed at Tokushima University. The pulsed power generator was composed of a B-PFN and a DC high-voltage power supply (ALE Model 102, Lambda-EMI, U.S.). The circuit constants $C_1$ and $L_1$ were 295 pF and 300 nH, respectively. The voltage and current of the output pulses were measured using a voltage probe (HVP-39pro, PINTEC, China) and current transformer (CURRENT MONITOR MODEL 110A, PEARSON ELECTRONICS, INC., U.S.), respectively, and the waveforms were monitored by an oscilloscope (DSO1024A, Agilent Technologies, U.S.). Under our experimental conditions, an electroporation cuvette with aluminum electrodes spaced 4 mm apart (Nepa Gene Co., Ltd., Japan) and filled with the cell suspension and silicon oil (Shin-Etsu Chemical Co., Ltd., Japan) resulted in an average pulse width at half maximum of approximately 70 ns (Fig 1A).

### Immunoblot analysis

Cells were lysed in RIPA buffer (50 mM Tris pH 7.5, 150 mM NaCl, 1 mM EDTA, 0.1% SDS, 1% NP-40, 0.5% deoxycholic acid) with protease inhibitor cocktail (Nacalai Tesque) and phosphatase inhibitor cocktail (Biotool). Immunoblot analysis was performed as previously described using Blocking One (Nacalai Tesque) or Blocking One-P (Nacalai Tesque) and WesternSure ECL Substrate (Li-Cor Biosciences). Protein was visualized by Ez-Capture II (ATTO Corp), and the band intensities were quantified using Image Studio software (LiCor Biosciences). The sources of antibodies were as follows: Phospho-Ser51-eIF2α (D9G8 #3398) (Cell Signaling Technology); eIF2α (D7D3 #5324) (Cell Signaling Technology); HRI (SC-30143) (Santa cruz); GAPDH (M171-3) (MBL); ATF4 (D4B8 #11815) (Cell Signaling Technology); ATF3 (SC-81189) (Santa cruz); CHOP (15204-1-AP) (Proteintech); XBP1s (D2C1F #12782) (Cell Signaling Technology); Ribophorin (Homemade).

### ROS production detection

At the end of the treatment schedule, cells were incubated with 10 µM $CM$-$H_2DCFDA$ (Thermo Fisher) in culture media for 30 min. Then, cells were washed with PBS, and the cell pellets collected by trypsinization were resuspended in 10% FBS-supplemented DMEM and analyzed for intracellular ROS production by flow cytometry S3e (Bio-Rad). All experiments were performed in three independent replicates.

### Cell viability assay

Cell viability was determined by WST-8 assay (Dojin Laboratory) according to the manufacturer's instructions. Briefly, WST-8 solution was added to cells in 96-well plates and the optical density of each well was read at 450 nm using a microplate reader EMax Plus (Molecular Devices) followed by incubation for 1, 2, and 4 h after nsPEF treatment.

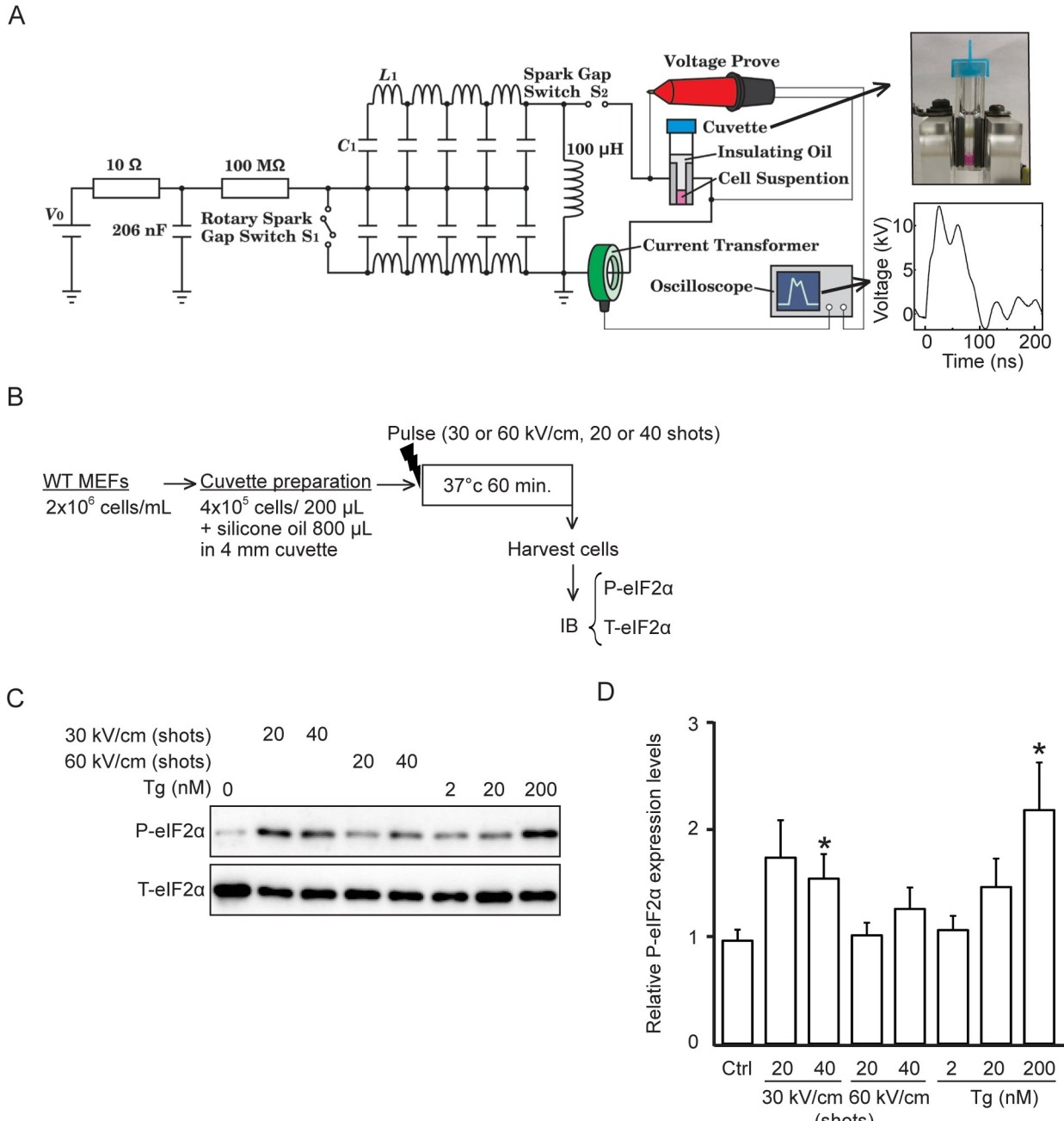

**Fig 1. Phosphorylation of eIF2α is induced in WT MEF cells by 40 shots of nsPEFs with 70-ns duration and 30-kV/cm electric fields.** (**A**) The circuit configuration of the B-PFN as an nsPEF generator. The right upper panel shows a photograph of the nsPEF delivery device with a 4-mm gap cuvette. The right lower panel shows typical waveforms of nsPEFs using a 4-mm gap cuvette. (**B**) Experimental protocol. Resuspended WT MEF cells (4 x 10⁵) were loaded into a 4-mm gap cuvette and covered with 800 μL silicone oil. After the indicated nsPEF treatment, WT MEF cells were collected into a 1.5-mL tube and incubated at 37°C for 1 h followed by immunoblot analysis. (**C**) Representative immunoblots of phosphorylated eIF2α and total eIF2α in WT MEF cells 1 h after the indicated nsPEF treatment. An ER stressor Tg served as a positive control for eIF2α phosphorylation. (**D**) Densitometry quantification of phosphorylated eIF2α normalized to the total eIF2α level in WT MEF cells 1 h after the indicated nsPEF treatment. Error bars show the means ± SEM ($n = 8$, $^*P < 0.05$).

## Mitochondrial membrane potential measurements

The changes in mitochondrial membrane potential were assayed using using the lipophilic cationic probe JC-1 (Setareh Biotech). The cells were incubated with 5 μg/mL JC-1 dye in culture media for 1 h, subsequently washed with PBS and then resuspended in PBS. The samples were then analyzed using, cells were removed probe, resuspended in PBS The emitted green (JC-1 monomer) and red (JC-1 polymer) fluorescence were detected by a fluorescence microscope (Olympus) and were analyzed for mitochondrial membrane potential using ImageJ (NIH).

## Statistical analysis

Statistical analysis was performed using Student's *t*-test. Data are expressed as the mean ± SEM. A difference was considered to be statistically significant when the *P* value was less than 0.05, unless otherwise stated.

## Results

### Phosphorylation of eIF2α is induced in the WT MEF cells by 40 shots of nsPEFs with 70-ns duration and 30-kV/cm electric fields

Previous studies have demonstrated that 80-ns PEFs induce eIF2α phosphorylation, which is a hallmark of the ISR, in HeLa S3 cells [23]. In this study, we applied 70-ns PEFs, which did not cause significant heat generation or cell death to MEFs (Fig 1A and S1A and S1B Fig). PEFs result in discharge on the surface of the cell suspension, causing insulation breakdown. To avoid this, we placed 800 μl of silicone oil on 200 μl of the cell suspension, which helped to increase the electric field (Fig 1A and 1B). To determine the optimal conditions of nsPEF treatment for eIF2α phosphorylation, we employed two pulse numbers (20 or 40 shots) and two electric fields (30 kV/cm or 60 kV/cm) (Fig 1B). Under these conditions, there were no physiologically meaningful temperature shifts in the cell suspension or significant cell damage (S1A and S1C Fig). eIF2α phosphorylation induction was enhanced by 30-kV/cm nsPEFs to a greater extent than by 60-kV/cm nsPEFs (Fig 1C and 1D). Both 40 shots of nsPEFs and 200 nM thapsigargin (Tg) treatment, which is a well-validated ISR activator, constantly induced eIF2α phosphorylation. Therefore, 40 shots of 30-kV/cm nsPEFs were used for further studies.

### HRI is responsible for eIF2α kinase ISR activation by 70-ns PEF treatment

To better understand the mechanism of eIF2α phosphorylation, we previously established 4KO cells, in which the four eIF2α kinase genes were deleted using CRISPR/Cas9-mediated genome editing, and single eIF2α kinase-rescued 4KO cell lines (single rescue 4KO cells), in which one of four eIF2α kinase genes was overexpressed in 4KO cells [22]. Because there is overlap in the type of stresses among each eIF2α kinase, the magnitude of the activation of the primary kinase overshadows that of the secondary kinase, making the latter kinase difficult to detect (Fig 2A). For overcoming this problem, single rescue 4KO cells are powerful tools for determining the eIF2α kinase responsible for ISR activation. We previously reported that the four known eIF2α kinases are sufficient for the ISR and that there are no additional eIF2α kinases in vertebrates. As expected, eIF2α phosphorylation was completely abolished when we applied nsPEFs to the 4KO cells (Fig 2B). We next used single rescue 4KO cells and found that the phosphorylation of eIF2α in response to nsPEFs was successfully recovered when HRI was expressed, indicating that of the eIF2α kinases, HRI is responsible for nsPEFs (Fig 2C). Activation of the ISR was further confirmed by induction of ISR downstream target genes such as ATF4 and ATF3 (S2D and S2E Fig). As we expected, proapoptotic factor CHOP and UPR marker XBP1s were not induced by nsPEF treatment, indicating that nsPEFs did not cause

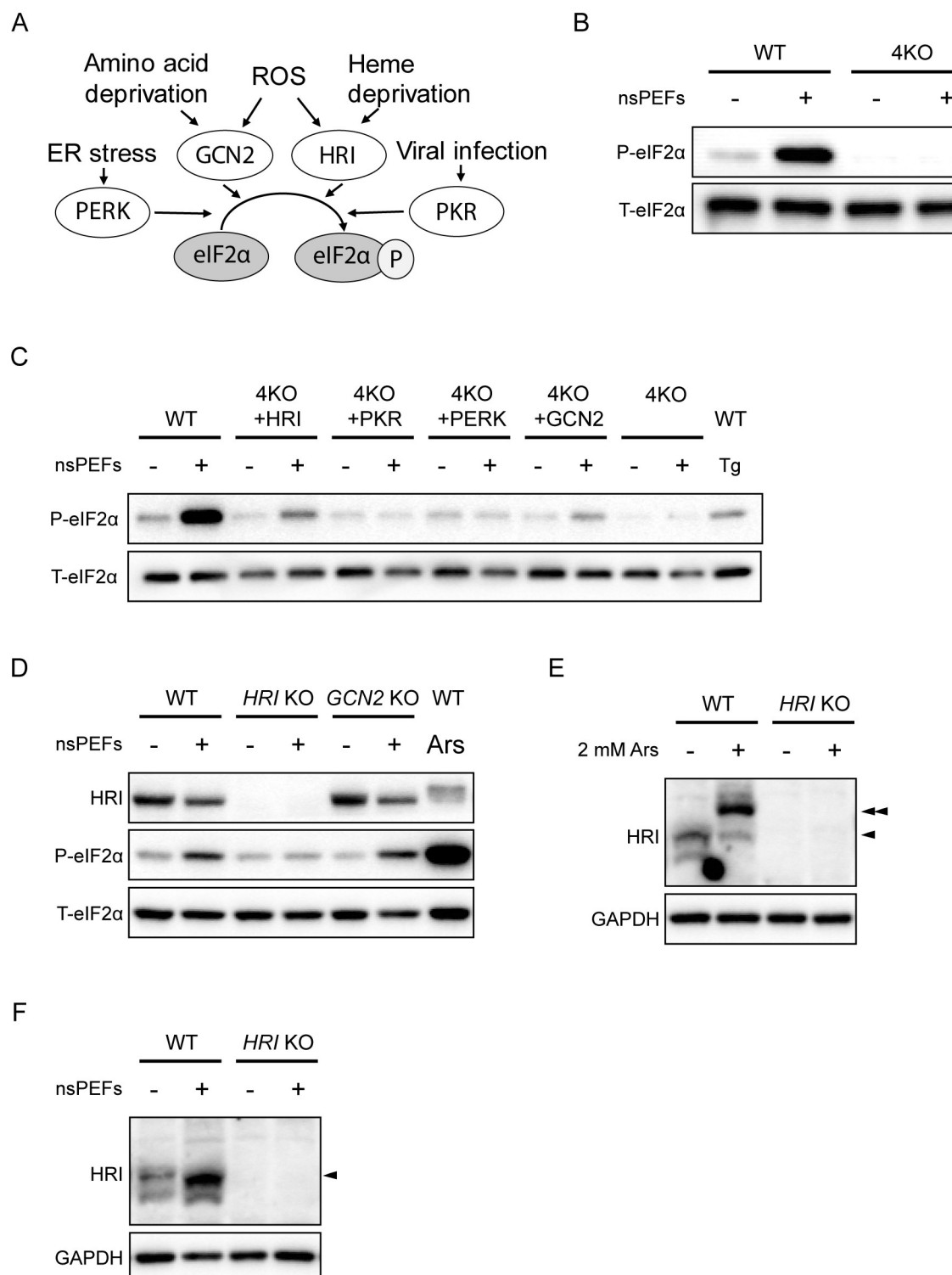

**Fig 2. The eIF2α kinase HRI is responsible for nsPEF-induced eIF2α phosphorylation.** (**A**) Various stressors phosphorylate eIF2α via activation of four eIF2α kinases, PERK, GCN2, HRI and PKR. (**B**) Representative immunoblots of phosphorylated eIF2α and total eIF2α in WT and 4KO MEF cells 1 h after nsPEF treatment (40 shots of 70-ns duration and 30-kV/cm electric fields). (**C**) Representative immunoblots of phosphorylated eIF2α and total eIF2α in WT, 4KO and single eIF2α kinase-rescued 4KO MEF cells 1 h after nsPEF treatment (40 shots of 70-ns duration and 30-kV/cm electric fields). Treatment with 200 nM Tg served as a positive control for eIF2α phosphorylation. (D) Representative immunoblots of HRI, phosphorylated eIF2α and total eIF2α in WT *HRI* KO and *GCN2* KO Hap1 cells 1 h after nsPEF treatment (40 shots of 70-ns duration and 30-kV/cm electric fields). nsPEFs induced eIF2α phosphorylation in WT

and *GCN2* KO Hap1, but did not induce eIF2α phosphorylation in *HRI* KO Hap1. (E) Representative Phos-tag immunoblots of HRI and immunoblots of GAPDH in WT and *HRI* KO Hap1 cells 1 h after 2 mM arsenite treatment. The arrowhead indicated the phosphorylated form of HRI. (F) Representative Phos-tag immunoblots of HRI and immunoblots of GAPDH in WT and *HRI* KO Hap1 cells 1 h after nsPEF treatment (40 shots of 70-ns duration and 30-kV/cm electric fields). The arrowheads indicated the phosphorylated form of HRI.

apoptosis or ER stress (S2D and S2E Fig). Overexpression of a single eIF2a kinase may not reflect normal physiological function. To exclude this possibility, we established single *HRI* KO and *GCN2* KO cells using CRISPR/Cas9 system in human Hap1 cell, respectively. Induction of phosphorylated eIF2α by nsPEFs was observed in WT and *GCN2* KO Hap1 cells, but not in *HRI* KO cells (Fig 2D). Furthermore, we confirmed that nsPEFs phosphorylated HRI in WT Hap1 cell by phos-tag SDS-PAGE as well as known a known HRI activator arsenite (Ars) did (Fig 2E and 2F, arrowheads). Thus, these data demonstrated that nsPEFs phosphorylates the eIF2α via HRI activation.

## ROS exposure activates the eIF2α kinase HRI to initiate ISR activation

Although HRI is well known to be activated by heme deficiency in immature erythroid cells [24], mRNA expression for HRI has been identified across a wide range of tissues [25]. ROS have been reported to act as an HRI activator [26] and HRI KO cells suffered from increased levels of ROS and apoptosis [27]. To verify previous reports, we analyzed eIF2α phosphorylation in single rescue 4KO cells using the ROS agent hydrogen peroxide ($H_2O_2$) and the ROS scavenger N-acetyl-L-cysteine (NAC). In the current study, eIF2α phosphorylation after $H_2O_2$ exposure was recovered in HRI-rescued 4KO cells (Fig 3A and 3B). Furthermore, blockade of ROS via administration of NAC led to reduced HRI-mediated eIF2α phosphorylation after $H_2O_2$ exposure (Fig 3A and 3B). Thus, these data demonstrated that ROS activate the ISR mainly by HRI activation.

## ROS-activated HRI phosphorylates eIF2α in response to 70-ns PEF treatment

nsPEF-induced HRI activation may be assumed to be mediated by ROS. Indeed, the generation of ROS by nsPEF treatment has been reported by several groups [28–30], but some contradicting results exist [31]. For example, 300-ns 45-kV/cm nsPEF to Jurkat cells [28], 100-ns 30-kV/cm nsPEF to BxPC-3 cells [29] and 100-ns 40-kV/cm nsPEF to B16f10 or Panc-1 cells [30] increased intracellular ROS. On the other hand, 300-ns and 40-, 50- or 60-kV/cm nsPEF to E4 squamous cells did not increase intracellular ROS [31]. To examine the contribution of ROS on nsPEF-induced HRI activation, we analyzed eIF2α phosphorylation in each single rescue 4KO cell line after 70-ns PEF treatment with or without NAC. As expected, eIF2α was significantly phosphorylated by nsPEF treatment in WT and HRI-rescued 4KO cells but not in PERK-, PKR- or GCN2-rescued 4KO cells (Fig 4A and 4B). NAC treatment decreased nsPEF-induced eIF2α phosphorylation in WT and HRI-rescued 4KO cells to almost basal levels, indicating that nsPEF-induced ROS are a main cause of nsPEF-induced HRI activation (Fig 4A and 4B). To further confirm the nsPEF-induced ROS generation, we monitored the intracellular ROS content using the CM-$H_2$DCFDA fluoroprobe, a membrane-permeable form of a ROS indicator (Fig 4C). nsPEF treatment elevated the ROS level in both WT and 4KO cells, and NAC attenuated ROS generation produced by nsPEFs (Fig 4C and S2C Fig). Mitochondria are considered as the main source of ROS in the cell. Therefore, we monitored mitochondrial membrane potential using JC-1 dye but the mitochondrial membrane potential was not found to be affected by nsPEFs treatment (S2A and S2B Fig). Altogether, our data strongly

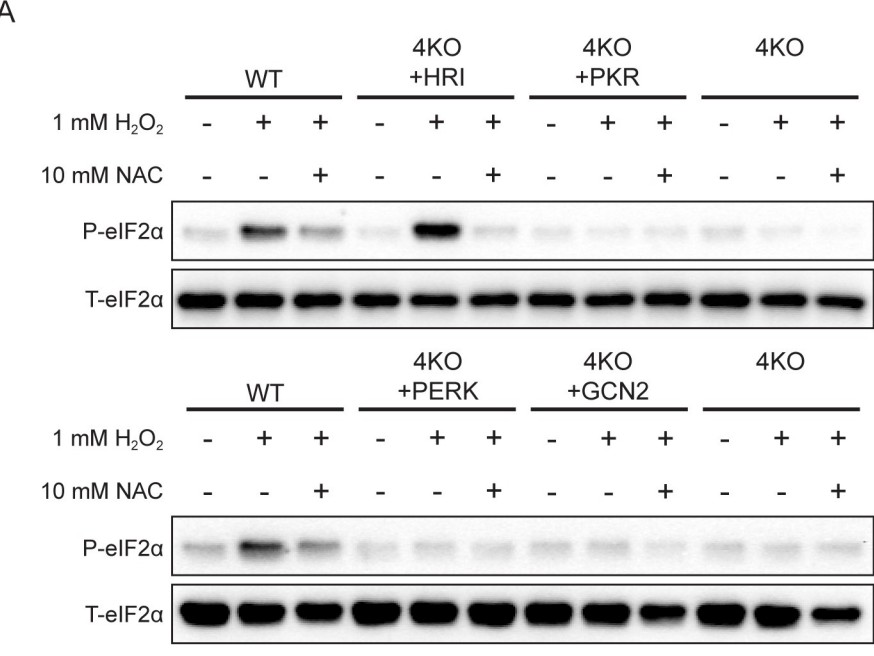

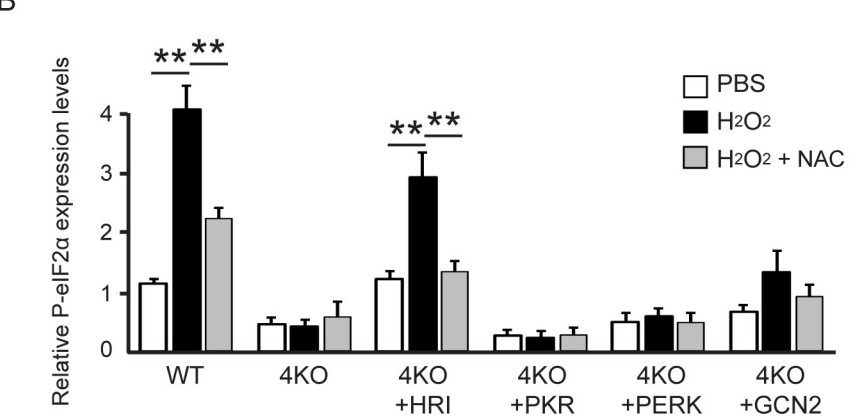

**Fig 3. The eIF2α kinase HRI is responsible for ROS-induced eIF2α phosphorylation.** (**A**) Representative immunoblots of phosphorylated eIF2α and total eIF2α in WT, 4KO and single eIF2α kinase-rescued 4KO MEF cells 1 h after 1 mM $H_2O_2$ treatment with or without 10 mM NAC. (**B**) Densitometry quantification of phosphorylated eIF2α normalized to the total eIF2α level in WT, 4KO and single eIF2α kinase-rescued 4KO MEF cells 1 h after 1 mM $H_2O_2$ treatment with or without 10 mM NAC. Error bars show the means ± SEM ($n$ = 4–9, **$P < 0.01$).

suggest that nsPEFs generate ROS, which increase eIF2α phosphorylation via HRI activation (Fig 4D).

## Discussion

Dysregulation of ISR signaling contributes to a wide range of pathological conditions linked to inflammation [6], diabetes [7], cancer [8], and neurodegenerative diseases [9]. Thus, modulation of the ISR may hold promise for a new therapeutic tool to treat various forms of human disease. Here, we present studies showing that 70-ns 30-kV/cm nsPEF treatment induced ISR activation in WT MEF cells. Using both 4KO cells and single rescue 4KO cells, we found that

A

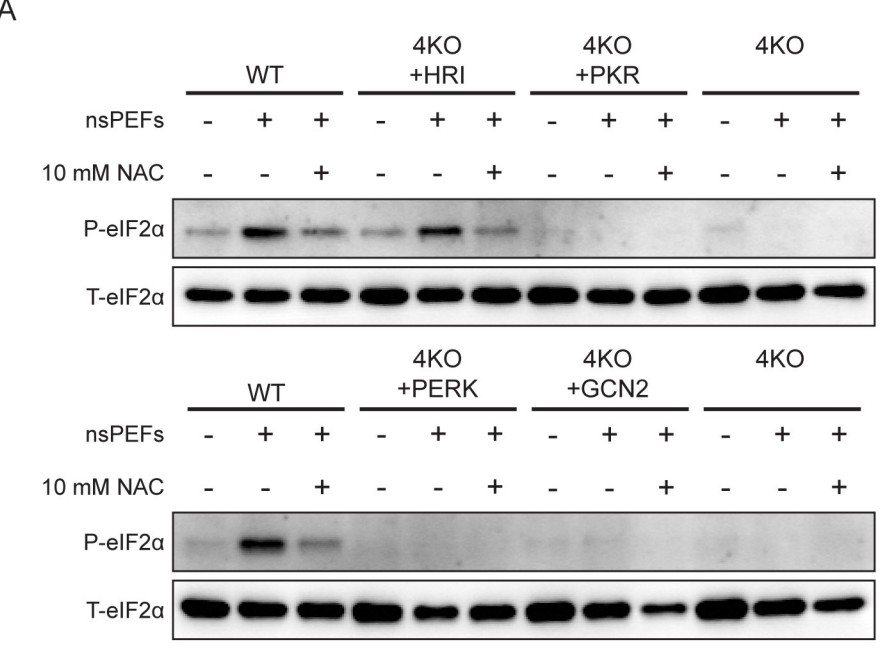

B

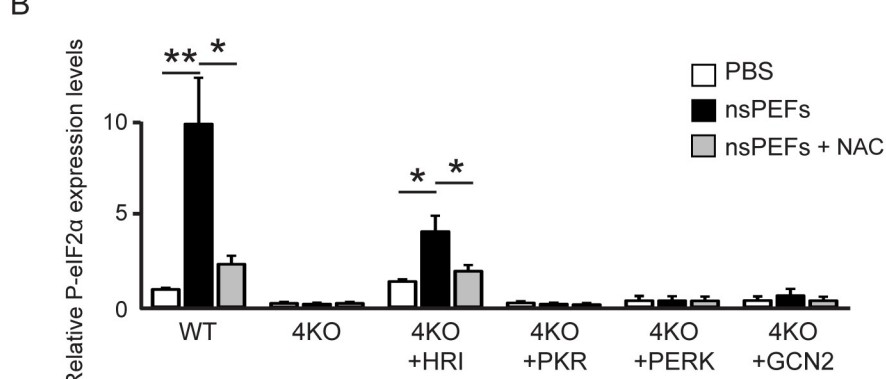

C

D

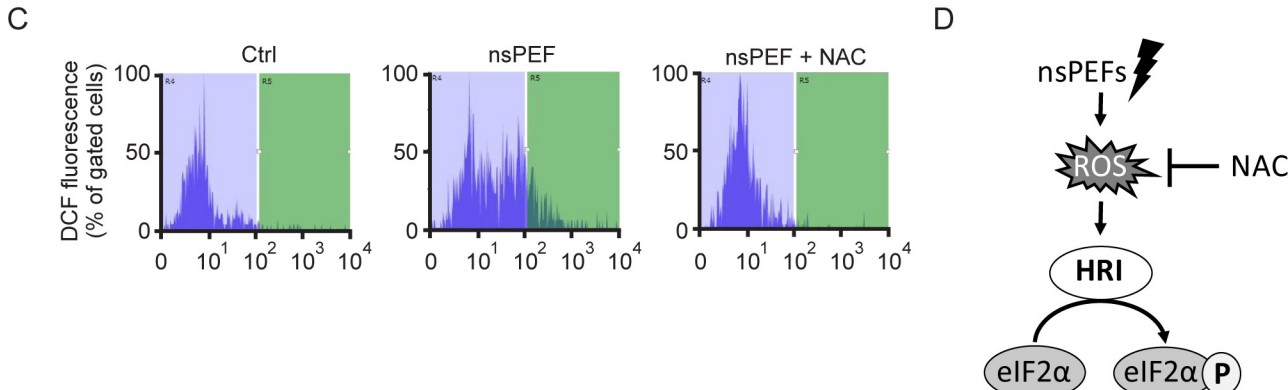

**Fig 4. ROS-activated HRI phosphorylates eIF2α during nsPEF treatment.** (**A**) Representative immunoblots of phosphorylated eIF2α and total eIF2α in WT, 4KO and single eIF2α kinase-rescued 4KO MEF cells 1 h after nsPEF treatment (40 shots of 70-ns duration and 30-kV/cm electric fields) with or without 10 mM NAC. (**B**) Densitometry quantification of phosphorylated eIF2α normalized to the total eIF2α level in WT, 4KO and single eIF2α

kinase-rescued 4KO MEF cells 1 h after nsPEF treatment (40 shots of 70-ns duration and 30-kV/cm electric fields) with or without 10 mM NAC. Error bars show the means ± SEM ($n$ = 5–7, $^*P < 0.05$, $^{**}P < 0.01$). (**C**) Representative flow cytometric profiles of intracellular ROS levels in nsPEF-treated WT MEFs with or without 10 mM NAC using the CM-H$_2$DCFDA fluoroprobe. (**D**) Proposed schematic model of the phosphorylation of eIF2$\alpha$ by nsPEF treatment. nsPEF treatment (40 shots of 70-ns duration and 30-kV/cm electric fields) generates intracellular ROS, which increases eIF2$\alpha$ phosphorylation via HRI activation.

HRI is activated by nsPEF treatment and that activated HRI phosphorylates eIF2$\alpha$. Furthermore, using the ROS indicator CM-H$_2$DCFDA and the ROS scavenger NAC, we demonstrated that nsPEF-generated ROS are required for HRI activation.

Mitochondria are considered the main source of ROS in the cell, and mitochondrial dysfunction is often associated with increased ROS production [32]. The distinct effect of nsPEFs from classical plasma membrane electroporation is the opening of nanopores into intercellular organelles, such as the mitochondria, nucleus and ER [19]. The nsPEF-induced release of cytochrome c from the mitochondria into the cytoplasm suggests that the mitochondria are intracellular targets for nsPEFs [33]. Indeed, nsPEF-induced permeabilization of mitochondrial membranes has been reported [34]. ROS production from mitochondria may depend on several factors, such as proton motive force or the redox state of the NADH pool, but the molecular mechanism of ROS production by nsPEFs remains to be elucidated.

nsPEF-triggered ROS production has been observed by several [28–30] groups, but one report indicated that nsPEFs have no or minimal effects on ROS production [31]. This discrepancy can be explained by cell-type differences. Pakhomova et al. reported that ROS production increased with time after treatment in nsPEF-sensitive Jurkat cells but remained stable in nsPEF-resistant U937 cells under the same conditions of 300-ns 45-kV/cm nsPEFs [28]. In the case of E4 squamous cells, ROS production was not observed under the same 300-ns duration with electric field strength from 0 to 60 kV/cm [31]. These results suggest that nsPEF-induced ROS production is cell-type dependent, probably due to different intercellular antioxidant activity or different mitochondrial vulnerability to nsPEFs.

However, different nsPEF conditions may lead to different cellular consequences. We previously reported that PERK and GCN2 are involved in eIF2$\alpha$ phosphorylation caused by 80-ns 20-kV/cm nsPEFs [23], demonstrating that nsPEF-induced eIF2$\alpha$ phosphorylation was reduced but not abolished in PERK/GCN2 double-KO MEFs, indicating that HRI is also involved in nsPEF-induced eIF2$\alpha$ phosphorylation. However, in the present study, we did not observe PERK or GCN2 activation in PERK- or GCN2-rescued 4KO MEF cells under 70-ns 40-kV/cm nsPEF treatment. Because PERK and GCN2 are known to be activated by ER stress [35], 80-ns 20-kV/cm nsPEF treatment may form nanopores in the ER membrane, as well as in the mitochondria membrane, thereby causing ER stress. Thus, we assume that intracellular membranes may be affected differentially by nsPEF conditions. Optimized nsPEF conditions such as pulse duration, rise time, pulse amplitude and number for organelle-specific effects are worthy of further exploration.

nsPEFs have attracted much attention during the last decade because they may represent a drug-free therapy. The therapeutic use of nsPEFs allows the manipulation of cell fate in two distinct ways: first, nsPEF treatment may lead to cell death; second, nsPEFs may enhance cellular function. For instance, nsPEF therapy has been proven effective in treating cancer in a mouse xenograft model such as melanoma [36], basal cell carcinoma [37] and pancreatic carcinoma [38]. nsPEFs initiate apoptosis in a nonthermal manner, presumably through mitochondrial and caspase-dependent mechanisms. However, nsPEFs regulate diverse biological effects, such as enhancement of chondrogenic differentiation [39], endothelial cell proliferation [40], and damage-free excitation of peripheral nerves [41] and cardiomyocytes [42]. The double-edged sword of nsPEF effects may depend on cell type or cell type-specific signaling pathways.

Activation of the ISR could also decide cell fate (either survival or apoptosis) depending on the magnitude of eIF2α phosphorylation. Our results indicated that the two-faced nature of nsPEFs is mediated in part through ISR activation. More detailed mechanism is needed to be investigated in near future.

## Supporting information

**S1 Fig. nsPEFs did not affect the medium temperature and cell proliferation.** (A) Measurements of medium temperature before and after nsPEFs treatment. Error bars show the means ± SEM ($n = 6$, $^*P < 0.05$). (B) Representative images and quantification of viability in WT or 4KO cells at 4 h with mock or nsPEF treatment. Cell viability was detected using WST-8 reagent, and the values are shown as the mean as the mean ± SEM.
(PDF)

**S2 Fig. nsPEFs induced the ISR activation without mitochondrial membrane potential.** (A) Representative images of JC-1 dye stained cells treated with nsPEF treatment (40 shots of 70-ns duration and 30-kV/cm electric fields) or a mitochondrial uncoupler FCCP used as positive control. Green fluorescence represents the monomeric form of JC-1, indicating dissipation of mitochondrial membrane potential. (B) Quantification of Green (JC-1 monomer)/Red (JC-1 polymer) fluorescence ratio in cells 1 h after nsPEF treatment (40 shots of 70-ns duration and 30-kV/cm electric fields) ($n = 30$, $^{**}P < 0.01$, $n.s.$ = not significant). (C) Quantification of intracellular ROS levels in nsPEF-treated WT or 4KO MEFs using the CM-H$_2$DCFDA fluoroprobe. Error bars show the means ± SEM ($n = 3$–5, $^*P < 0.05$, $n.s.$ = not significant). (D) Representative immunoblots of ATF4, ATF3, CHOP, XBP1s, and Ribophorin 1 h after treatment with the nsPEFs and 2 μg/mL Tm in WT Hap1 cells. (E) Densitometry quantification of ATF4, ATF3, CHOP, and XBP1s expression were normalized to the Ribophorin expression as the mean + SEM ($n = 3$, $^*P < 0.05$, $^{**}P < 0.01$, $n.s.$ = not significant).
(PDF)

**S1 Data.**
(PPTX)

## Acknowledgments

We thank C. Kimura (Tokushima University) for help with manuscript preparation and Oyadomari laboratory members for helpful discussion.

## Author Contributions

**Conceptualization:** Seiichi Oyadomari.

**Data curation:** Miho Oyadomari.

**Formal analysis:** Yoshimasa Hamada, Seiichi Oyadomari.

**Funding acquisition:** Seiichi Oyadomari.

**Investigation:** Yoshimasa Hamada, Yuji Furumoto, Akira Izutani, Shusuke Taniuchi, Masato Miyake.

**Methodology:** Yuji Furumoto, Akira Izutani, Kenji Teranishi, Naoyuki Shimomura, Seiichi Oyadomari.

**Project administration:** Miho Oyadomari, Seiichi Oyadomari.

**Resources:** Shusuke Taniuchi.

**Supervision:** Masato Miyake, Naoyuki Shimomura, Seiichi Oyadomari.

**Writing – original draft:** Yoshimasa Hamada, Seiichi Oyadomari.

**Writing – review & editing:** Seiichi Oyadomari.

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
