## [Decision Letter · Decision Letter 0]

3 Sep 2019

PONE-D-19-21632

Nanosecond pulsed electric fields induce the integrated stress response via reactive oxygen species-mediated heme-regulated inhibitor (HRI) activation

PLOS ONE

Dear Dr. Oyadomari,

Your manuscript has been reviewed by three experts in the field and their comments follow. They have raised a number of substantive concerns, which preclude acceptance of your paper. For the paper to be reconsidered, all concerns raised by the three reviewers have to be addressed in full and satisfactorily. The concern about using eIF2a kinase overexpression only should be addressed by performing new experiments or adding new results.  

We would appreciate receiving your revised manuscript within four months. To enhance the reproducibility of your results, we recommend that if applicable you deposit your laboratory protocols in protocols.io, where a protocol can be assigned its own identifier (DOI) such that it can be cited independently in the future. For instructions see: http://journals.plos.org/plosone/s/submission-guidelines#loc-laboratory-protocols

We look forward to receiving your revised manuscript.

Kind regards,

Dong-Yan Jin

Academic Editor

PLOS ONE

Journal Requirements:

1. PLOS ONE now requires that authors provide the original uncropped and unadjusted images underlying all blot or gel results reported in a submission’s figures or Supporting Information files. This policy and the journal’s other requirements for blot/gel reporting and figure preparation are described in detail at https://journals.plos.org/plosone/s/figures#loc-blot-and-gel-reporting-requirements and https://journals.plos.org/plosone/s/figures#loc-preparing-figures-from-image-files. When you submit your revised manuscript, please ensure that your figures adhere fully to these guidelines and provide the original underlying images for all blot or gel data reported in your submission. See the following link for instructions on providing the original image data: https://journals.plos.org/plosone/s/figures#loc-original-images-for-blots-and-gels.

Additional Editor Comments:

One reviewer recommended rejection based on a major concern about overexpression. The other two reviewers also made specific suggestions. You should address their comments carefully and thoroughly and your paper will be re-reviewed by the same reviewers if they are available.

Reviewers' comments:

Reviewer's Responses to Questions

**Comments to the Author**

1. Is the manuscript technically sound, and do the data support the conclusions?

Reviewer #1: Partly

Reviewer #2: Partly

Reviewer #3: Yes

2. Has the statistical analysis been performed appropriately and rigorously? 

Reviewer #1: No

Reviewer #2: Yes

Reviewer #3: N/A

3. Have the authors made all data underlying the findings in their manuscript fully available?

Reviewer #1: No

Reviewer #2: No

Reviewer #3: Yes

4. Is the manuscript presented in an intelligible fashion and written in standard English?

Reviewer #1: Yes

Reviewer #2: Yes

Reviewer #3: Yes

5. Review Comments to the Author

Reviewer #1: The manuscript by Hamada et al described the study of nanosecond pulsed electric fields (nsPEFs) on integrated stress response (ISR). They concluded that HRI is the eIF2alpha kinase responsible for the ISR activation by nsPEF-mediated oxidative stress. However, the scope of manuscript is very limited and preliminary. The second arem of ATF4 induction was not investigated in this manuscript. Their study did not increase our knowledge on the mechanisms as how and why nsPEFs may be useful in clinical practice. There is also a concern on the sole approach of using overexpression of single eIF2a kinase. Studies using single KO of each kinase shall be included, if indeed HRI is the sole eIF2a kinase responsible. It is important to note that overexpression of single eIF2a kinase may not be physiological due to the high level of kinase achieved.

The specific comments are:

1. Citations on the literature of HRI are inadequate; missing several important papers on the activation of HRI by oxidative stress and protection of cells against oxidative insults.

2. Fig. 1C, Fig 4A and B

Why is eIF2aP in 4KO+HRI so much lower than WT?

It seems that GCN2 also contributes to increased eIF2aP upon nsPEFs.

3. Fig. 2A, Similarly to Fig. 1C, why is Why is eIF2aP in 4KO+HRI so much lower than WT upon H2O2 treatment.

4. Fig. 4C, The FACS plots for ROS measurement shall be presented

Reviewer #2: The integrated stress response (ISR), a mechanism by which cells modulate protein synthesis in response to cellular stress, is controlled by the phosphorylation of translation initiation factor eIF2alpha. Four eIF2 kinases (HRI, GCN2, PERK, and PKR) each sense and are activated in response to different cellular stresses, and integrate on the shared eIF2alpha substrate. HRI, for instance, has been reported to be activated by elevated ROS in multiple cells types and heme deprivation in erythroid cells. In this paper, the authors utilize previously engineered MEF cells, which express each eIF2 kinase individually, to identify the kinase responsible for eIF2alpha phosphorylation in response to nsPEFs, a potential therapeutic tool for ISR activation. The authors report that 30 kV/cm nsPEFs promote the generation of ROS, and that cells expressing HRI as the sole eIF2 kinase display with heightened levels of eIF2alpha phosphorylation. Moreover, reduction of ROS levels by addition of a ROS scavenger, decreases eIF2alpha phosphorylation in the HRI expressing cells. Taken together, these results suggest that HRI is the eIF2 kinase activated in response to 30 kV/cm nsPEFs and reveal that the method of HRI activation in these conditions relies upon elevated ROS levels. It remains unclear, however, if HRI performs this function in a wild-type state in which all four eIF2 kinases are expressed at their endogenous levels, or only in these single kinase overexpression cell lines.

In general, the results are convincing and well described. I have a few comments that the authors should address to strengthen the conclusions of the manuscript.

1. Page 3, line 71: reference 11 does not support the statement that “enhancement of ISR signaling has been suggested to have beneficial effects,” and should be removed as a reference from this sentence as it is written.

2. Figures 1D, 3B, 4B, and 4C: why were SEM recorded rather than SD? The Materials and Methods indicate that data are represented as the mean +/- SD.

3. Page 8, line 153 and page 8, lines 158-160. What data support the two statements that PEFs did not result in meaningful temperature shifts or changes in cell viability?

4. Figure 3: The immunoblot for eIF2alpha phosphorylation in the 4KO+GCN2 cell line does not reflect the quantification provided. Is there another image that better represents the quantification?

5. Are ROS levels elevated to similar levels in response to nsPEFs in the WT, 4KO, 4KO+HRI, 4KO+PKR, 4KO+PERK, and 4KO+GCN2 cells lines?

6. The main conclusion of the manuscript is that nsPEFs generate ROS that activate HRI, leading to eIF2alpha phosphorylation. However, HRI activation under these conditions has not been demonstrated. A direct demonstration of activation/auto-phosphorylation of HRI would strengthen the conclusion.

7. While the data clearly indicate that eIF2alpha phosphorylation is induced to the greatest extent in the 4KO+HRI overexpression cell line (as compared to the other eIF2 kinase rescue lines), it is unclear to me if HRI serves to phosphorylate eIF2 in response to nsPEFs in the natural cellular state (i.e. when HRI and the other eIF2 kinases are expressed together at their endogenous levels). One strategy to asses this would be the individual knockdown or knockout of HRI from the WT cell line, followed by the assessment of eIF2alpha phosphorylation in response to PEFs.

Reviewer #3: In this manuscript the authors show that nanosecond pulsed electric fields (nsPEFs) activate the integrated stress response in mouse embryo fibroblasts (MEFs) via ROS-dependent HRI activation. The results are convincing and interesting, but the story is not quite complete. The ISR is usually activated by unfolded proteins in the cytosol, ER, mitochondria, or nucleus. These unfolded proteins can be induced by ROS, so it seems likely that ROS trigger HRI and eIF2alpha phosphorylation via unfolded proteins. The authors should present data to support or refute this idea. Detailed comments follow.

1. What happens to the wt MEFs after they are exposed to this dose of nsPEF? To the 4KO MEFs?

2. Is it possible that the molecular mechanism(s) that couple nsPEF to the ISR will be different in different cell types? (Do the authors have any evidence for this?)

3. The authors previously implicated PERK and GCN2 in nsPEF-induced ISR. Here they suggest that the discrepancy between that conclusion and the one defended here has to do with dose. Do they have direct evidence for this?

4. Does this dose of nsPEF cause cytochrome c release? A decrease in mitochondrial membrane potential?

6. PLOS authors have the option to publish the peer review history of their article (what does this mean?). If published, this will include your full peer review and any attached files.

Reviewer #1: No

Reviewer #2: No

Reviewer #3: Yes: David J. McConkey

---

## [Author Response · Author response to Decision Letter 0]

21 Jan 2020

We are grateful to the Editor and the reviewers for their critical comments and useful suggestions that have helped us to improve our paper. As indicated in the point-by-point responses below, we have taken all these comments and suggestions into account in the revised version of our manuscript.

We hope that the revised version of our paper is now suitable for publication in PLoS One and we look forward to hearing from you at your earliest convenience.

Reviewer #1

Major comments are:

1. The manuscript by Hamada et al described the study of nanosecond pulsed electric fields (nsPEFs) on integrated stress response (ISR). They concluded that HRI is the eIF2alpha kinase responsible for the ISR activation by nsPEF-mediated oxidative stress. However, the scope of manuscript is very limited and preliminary. The second arm of ATF4 induction was not investigated in this manuscript.

We confirmed the ATF4 induction in Hap1 cells with statistical significance (Sup. Fig. 2D and E). 

2. Their study did not increase our knowledge on the mechanisms as how and why nsPEFs may be useful in clinical practice. 

As this reviewer pointed out, we omitted the usefulness of nsPEFs in clinical practice in order to focus on mode of mechanism of nsPEFs in this paper.

3. There is also a concern on the sole approach of using overexpression of single eIF2a kinase. Studies using single KO of each kinase shall be included, if indeed HRI is the sole eIF2a kinase responsible. It is important to note that overexpression of single eIF2a kinase may not be physiological due to the high level of kinase achieved. 

We established HRI KO and GCN2 KO in Hap1 cells using CRISPR/Cas9 system, and we can confirm the HRI is responsible kinase for eIF2α phosphorylation by nsPEFs exposure (Fig. 2D-F). 

The specific comments are:

4. Citations on the literature of HRI are inadequate; missing several important papers on the activation of HRI by oxidative stress and protection of cells against oxidative insults.

According to the reviewer’s suggestions, we cited two important paper demonstrating that an essential role of HRI for red blood cells survival in heme deficiency (PMID:15931390) and HRI activation by oxidative stress (PMID: 22498744).

5. In Fig 4A and B, why is eIF2aP in 4KO+HRI so much lower than WT? It seems that GCN2 also contributes to increased eIF2aP upon nsPEFs.

We assumed that this lower phosphorylation level could be caused by lower expression level of HRI compared with that of WT. Unfortunately, we can’t validate the level of HRI overexpression compared with endogenous HRI due to lack of proper commercially available antibody to detect mouse HRI.

6. In Fig. 2A, Similarly to Fig. 1C, why is Why is eIF2aP in 4KO+HRI so much lower than WT upon H2O2 treatment.

It is the same reason above mentioned.

7. In Fig. 4C, The FACS plots for ROS measurement shall be presented.

We included the representative FACS plot for ROS measurement (Fig. 4C).

 

Reviewer #2: 

Major comments are:

The integrated stress response (ISR), a mechanism by which cells modulate protein synthesis in response to cellular stress, is controlled by the phosphorylation of translation initiation factor eIF2alpha. Four eIF2 kinases (HRI, GCN2, PERK, and PKR) each sense and are activated in response to different cellular stresses, and integrate on the shared eIF2alpha substrate. HRI, for instance, has been reported to be activated by elevated ROS in multiple cells types and heme deprivation in erythroid cells. In this paper, the authors utilize previously engineered MEF cells, which express each eIF2 kinase individually, to identify the kinase responsible for eIF2alpha phosphorylation in response to nsPEFs, a potential therapeutic tool for ISR activation. The authors report that 30 kV/cm nsPEFs promote the generation of ROS, and that cells expressing HRI as the sole eIF2 kinase display with heightened levels of eIF2alpha phosphorylation. Moreover, reduction of ROS levels by addition of a ROS scavenger, decreases eIF2alpha phosphorylation in the HRI expressing cells. Taken together, these results suggest that HRI is the eIF2 kinase activated in response to 30 kV/cm nsPEFs and reveal that the method of HRI activation in these conditions relies upon elevated ROS levels. It remains unclear, however, if HRI performs this function in a wild-type state in which all four eIF2 kinases are expressed at their endogenous levels, or only in these single kinase overexpression cell lines. In general, the results are convincing and well described. I have a few comments that the authors should address to strengthen the conclusions of the manuscript.

We thank this reviewer for appreciating our paper properly.

The specific comments are:

1. Page 3, line 71: reference 11 does not support the statement that “enhancement of ISR signaling has been suggested to have beneficial effects,” and should be removed as a reference from this sentence as it is written.

As this reviewer pointed out, we omitted the reference.

2. Figures 1D, 3B, 4B, and 4C: why were SEM recorded rather than SD? The Materials and Methods indicate that data are represented as the mean +/- SD.

We unified the statistical analysis into SEM.

3. Page 8, line 153 and page 8, lines 158-160. What data support the two statements that PEFs did not result in meaningful temperature shifts or changes in cell viability?

According to the reviewer’s suggestions, we monitored the cell viability (Sup. Fig. 1B）and the temperature shifts（Sup. Fig. 1A）after nsPEFs treatment. There is no adverse effect of nsPEFs on both cell viability and cell culture temperature. 

4. Figure 3: The immunoblot for eIF2alpha phosphorylation in the 4KO+GCN2 cell line does not reflect the quantification provided. Is there another image that better represents the quantification?

We performed the immunoblot analysis again and the representative image is now shown (Fig. 3A and B).

5. Are ROS levels elevated to similar levels in response to nsPEFs in the WT, 4KO, 4KO+HRI, 4KO+PKR, 4KO+PERK, and 4KO+GCN2 cells lines?

According to the reviewer’s suggestions, we measured the ROS levels again, and the level of ROS was elevated even in 4KO cells（Sup. Fig. 2C）

6. The main conclusion of the manuscript is that nsPEFs generate ROS that activate HRI, leading to eIF2alpha phosphorylation. However, HRI activation under these conditions has not been demonstrated. A direct demonstration of activation/auto-phosphorylation of HRI would strengthen the conclusion.

According to the reviewer’s suggestions, we performed phos-tag analysis of HRI. The HRI phosphorylation was induced by nsPEFs treatment as well as arsenite treatment (Fig. 2E and F).

7. While the data clearly indicate that eIF2alpha phosphorylation is induced to the greatest extent in the 4KO+HRI overexpression cell line (as compared to the other eIF2 kinase rescue lines), it is unclear to me if HRI serves to phosphorylate eIF2 in response to nsPEFs in the natural cellular state (i.e. when HRI and the other eIF2 kinases are expressed together at their endogenous levels). One strategy to asses this would be the individual knockdown or knockout of HRI from the WT cell line, followed by the assessment of eIF2alpha phosphorylation in response to PEFs.

According to the reviewer’s suggestions, we established HRI KO and GCN2 KO in Hap1 cells using CRISPR/Cas9 system, and we can confirm the HRI is responsible kinase for eIF2α phosphorylation by nsPEFs exposure (Fig. 2 D).

 

Reviewer #3: 

Major comments are:

In this manuscript the authors show that nanosecond pulsed electric fields (nsPEFs) activate the integrated stress response in mouse embryo fibroblasts (MEFs) via ROS-dependent HRI activation. The results are convincing and interesting, but the story is not quite complete. The ISR is usually activated by unfolded proteins in the cytosol, ER, mitochondria, or nucleus. These unfolded proteins can be induced by ROS, so it seems likely that ROS trigger HRI and eIF2alpha phosphorylation via unfolded proteins. The authors should present data to support or refute this idea. Detailed comments follow.

We thank this reviewer for appreciating our paper properly.

The specific comments are:

1. What happens to the wt MEFs after they are exposed to this dose of nsPEF? To the 4KO MEFs? 

We monitored the cell viability (Sup. Fig. 1B）and the cell culture medium temperature（Sup. Fig. 1A）under our condition of nsPEFs treatment. There is no significant effect of nsPEFs on both the cell viability and the cell culture medium temperature after the nsPEFs treatment.

2. Is it possible that the molecular mechanism(s) that couple nsPEF to the ISR will be different in different cell types? (Do the authors have any evidence for this?)

To exclude the possibility that our findings were specific to mouse fibroblasts, we examine the effect of nsPEFs in human Hap1 cells. The HRI-mediated ISR activation was induced also in Hap1 cells by the nsPEFs treatment, showing that ROS-mediated HRI activation is at least common in mouse fibroblasts and human Hap1 cells (Fig. 2 D-F, Sup. Fig. 2 D and E).

3. The authors previously implicated PERK and GCN2 in nsPEF-induced ISR. Here they suggest that the discrepancy between that conclusion and the one defended here has to do with dose. Do they have direct evidence for this?

Together with the above-mentioned point, we appreciate the thoughtful comments from this reviewer. Consistent with the previous our reports, nsPEFs induced the ISR. However, several factors such as s duration, number of pulses and intensity could lead to different biological processes. This could be attributed to varied phospholipid composition of organelle membrane among the different cell type, and this is an interesting question but beyond the limits of the present paper.

4. 4. Does this dose of nsPEF cause cytochrome c release? A decrease in mitochondrial membrane potential?

Change in mitochondrial membrane potential is known as a critical upstream signal of cytochrome C release. Therefore, we measured the mitochondrial membrane potential using JC-1 dye. We found that there was no significant change of mitochondrial membrane potential under our condition of nsPEFs treatment (Sup. Fig. 2A).

---

## [Decision Letter · Decision Letter 1]

10 Feb 2020

PONE-D-19-21632R1

Nanosecond pulsed electric fields induce the integrated stress response via reactive oxygen species-mediated heme-regulated inhibitor (HRI) activation

PLOS ONE

Dear Dr. Oyadomari,

Your revised paper was re-reviewed and the reviewers are satisfied with the revisions made. Your paper is therefore acceptable in principle. However, Reviewer 1 has remaining concerns that you should address in a further revised paper.

After careful consideration, we feel that your paper has merit but does not fully meet PLOS ONE’s publication criteria as it currently stands. Therefore, we invite you to submit a revised version of the manuscript that addresses the points raised during the review process.

We would appreciate receiving your revised manuscript by Mar 26 2020 11:59PM. To enhance the reproducibility of your results, we recommend that if applicable you deposit your laboratory protocols in protocols.io, where a protocol can be assigned its own identifier (DOI) such that it can be cited independently in the future. For instructions see: http://journals.plos.org/plosone/s/submission-guidelines#loc-laboratory-protocols

We look forward to receiving your revised manuscript.

Kind regards,

Dong-Yan Jin

Academic Editor

PLOS ONE

Reviewers' comments:

Reviewer's Responses to Questions

**Comments to the Author**

1. If the authors have adequately addressed your comments raised in a previous round of review and you feel that this manuscript is now acceptable for publication, you may indicate that here to bypass the “Comments to the Author” section, enter your conflict of interest statement in the “Confidential to Editor” section, and submit your "Accept" recommendation.

Reviewer #1: All comments have been addressed

Reviewer #2: All comments have been addressed

2. Is the manuscript technically sound, and do the data support the conclusions?

Reviewer #1: Yes

Reviewer #2: (No Response)

3. Has the statistical analysis been performed appropriately and rigorously? 

Reviewer #1: Yes

Reviewer #2: (No Response)

4. Have the authors made all data underlying the findings in their manuscript fully available?

Reviewer #1: Yes

Reviewer #2: (No Response)

5. Is the manuscript presented in an intelligible fashion and written in standard English?

Reviewer #1: Yes

Reviewer #2: (No Response)

6. Review Comments to the Author

Reviewer #1: The revised manuscript by Hamada et al has addressed all the concerns from the previous review. In particular, the single KO cell lines of HRI and GCN2 were generated and studied. In Furthermore, Activation of HRI in vivo in the Wt Hap1 cells was also demonstrated by phos-tag immunoblot analysis. One minor comment is that T-HRI band in Fig. 2E and F shall be unphophorylated HRI not total HRI. So, it shall be changed to HRI.

Reviewer #2: (No Response)

7. PLOS authors have the option to publish the peer review history of their article (what does this mean?). If published, this will include your full peer review and any attached files.

Reviewer #1: No

Reviewer #2: No

---

## [Author Response · Author response to Decision Letter 1]

18 Feb 2020

We agreed with the reviewer 1. The phosphorylation states of HRI on Phos-tag SDS-PAGE are now indicated on the right side of the blot with arrowheads. We thank the reviewer for bringing it to our attention.

---

## [Editor Report · Decision Letter 2]

19 Feb 2020

Nanosecond pulsed electric fields induce the integrated stress response via reactive oxygen species-mediated heme-regulated inhibitor (HRI) activation

PONE-D-19-21632R2

Dear Dr. Oyadomari,

We are pleased to inform you that your manuscript has been judged scientifically suitable for publication and will be formally accepted for publication once it complies with all outstanding technical requirements.

With kind regards,

Dong-Yan Jin

Academic Editor

PLOS ONE
---

## [Editor Report · Acceptance letter]

25 Feb 2020

PONE-D-19-21632R2 

Nanosecond pulsed electric fields induce the integrated stress response via reactive oxygen species-mediated heme-regulated inhibitor (HRI) activation 

Dear Dr. Oyadomari:

I am pleased to inform you that your manuscript has been deemed suitable for publication in PLOS ONE. Congratulations! Your manuscript is now with our production department. 

With kind regards,

on behalf of

Professor Dong-Yan Jin 

Academic Editor

PLOS ONE